

# Modulation of corticomotor excitability in response to distal focal cooling

Yekta Ansari[1,*], Anthony Remaud[2] and François Tremblay[1,2,*]

[1] School of Rehabilitation Sciences, Faculty of Heath Sciences, University of Ottawa, Ottawa, ON, Canada
[2] Clinical Neuroscience Lab, Bruyère Research Institute, Ottawa, ON, Canada
* These authors contributed equally to this work.

Corresponding author
François Tremblay,
ftrembla@uottawa.ca

## ABSTRACT

**Background:** Thermal stimulation has been proposed as a modality to facilitate motor recovery in neurological populations, such as stroke. Recently (*Ansari, Remaud & Tremblay, 2018*), we showed that application of cold or warm stimuli distally to a single digit produced a variable and short lasting modulation in corticomotor excitability. Here, our goal was to extend these observations to determine whether an increase in stimulation area could elicit more consistent modulation.

**Methods:** Participants ($n = 22$) consisted of a subset who participated in our initial study. Participants were asked to come for a second testing session where the thermal protocol was repeated but with extending the stimulation area from single-digit (SD) to multi-digits (MD, four fingers, no thumb). As in the first session, skin temperature and motor evoked potentials (MEPs) elicited with transcranial magnetic stimulation were measured at baseline (BL, neutral gel pack at 22 °C), at 1 min during the cooling application (pre-cooled 10 °C gel pack) and 5 and 10 min post-cooling (PC5 and PC10). The analysis combined the data obtained previously with single-SD cooling (*Ansari, Remaud & Tremblay, 2018*) with those obtained here for MD cooling.

**Results:** At BL, participants exhibited comparable measures of resting corticomotor excitability between testing sessions. MD cooling induced similar reductions in skin temperature as those recorded with SD cooling with a peak decline at C1 of respectively, −11.0 and −10.3 °C. For MEPs, the primary analysis revealed no main effect attributable to the stimulation area. A secondary analysis of individual responses to MD cooling revealed that half of the participants exhibited delayed MEP facilitation (11/22), while the other half showed delayed inhibition (10/22); which was sustained in the post-cooling phase. More importantly, a correlation between variations in MEP amplitude recorded during the SD cooling session with those recorded in the second session with MD cooling, revealed a very good degree of correspondence between the two at the individual level.

**Conclusion:** These results indicate that increasing the cooling area in the distal hand, while still eliciting variable responses, did produce more sustained modulation in MEP amplitude in the post-cooling phase. Our results also highlight that responses to cooling in terms of either depression or facilitation of corticomotor excitability tend to be fairly consistent in a given individual with repeated applications.

## INTRODUCTION

In recent years, the use of peripheral stimulation has gained a renewed attention as a potential adjuvant intervention in stroke rehabilitation. The use of thermal stimuli, in particular, has been proposed as a simple intervention to provide sensory stimulation to elicit motor facilitation in the affected arm or leg (*Chen, Liang & Shaw, 2005*; *Chen et al., 2011*). While there is clinical evidence that thermal stimulation (TS) through repeated applications of either cold or hot stimuli to the skin can assist in facilitating motor recovery in patients (*Hsu et al., 2013*; *Liang et al., 2012*), there is still very limited information regarding the neural basis underlying such facilitatory effects. For the proponents of this approach (*Hsu et al., 2013*; *Tai et al., 2014*), the facilitation was likely a reflection of the ability of thermal stimuli to elicit activation in both somatosensory and motor areas at the cortical level (*Davis et al., 1998*; *Gelnar et al., 1999*); leading to motor reorganization. To test this hypothesis, *Tai et al. (2014)* used transcranial magnetic stimulation (TMS) to probe changes in the motor maps in chronic stroke patients in response to either noxious (heat 46 °C, cold 7 °C) or innocuous (heat 41 °C, cold 20 °C) temperature stimuli targeting the affected upper extremity. Their results revealed a larger expansion of the motor maps in the lesioned hemisphere with noxious temperatures than with innocuous temperatures, although the magnitude of the effects was quite variable between patients. To further address the issue, we recently showed (*Ansari, Remaud & Tremblay, 2018*), also using TMS, that distal focal TS in the form of either innocuous cooling (10 °C) or warming (45 °C) produced a variable and short-lasting modulation in motor evoked potential (MEP) amplitude, irrespective of the age (young and old) and sex (men and women) of healthy participants. In fact, participants exhibited mixed patterns of modulation characterized by either depression or facilitation and only during actual cooling or warming stimulation. Our results also revealed that cooling was more likely to elicit modulation than warming; a finding consistent with the greater sensitivity to cold stimulation in human observers (*Jones & Ho, 2008*) and the larger effects reported for local cooling stimuli at the neurophysiological level (*Chang, Arendt-Nielsen & Chen, 2005*; *Dewhurst et al., 2005*).

One possibility to explain the variability we observed in response to TS is related to the depth and extent of the focal thermal effects at the peripheral level. Given the critical role of spatial summation in thermal sensibility (*Jones & Ho, 2008*; *Stevens, 2013*), it is possible that our stimulation, which was restricted to a single digit, might have been suboptimal to elicit modulation in corticomotor excitability, as reflected in MEP amplitude. Indeed, considerable spatial summation has been reported for both heat and cold modalities with increased area of stimulation leading to higher magnitude of sensation and lower detection threshold (reviewed in *Stevens, 2013*). Spatial summation appears to be particularly important for cold sensations given the higher density of innervation of cold receptors in the skin (*Stevens, 2013*). As demonstrated by *Stevens & Marks (1979)*, for a given degree of skin cooling, one can double the magnitude of cold sensation just by doubling the stimulated area. Thus, the area of stimulation in relation to spatial summation seems to be a critical factor when considering the central effects of thermal stimuli at the periphery.

In the present report, our goal was to extend our previous observations regarding the influence of distal focal TS on corticomotor excitability to determine whether extending the area of stimulation would lead to more consistent effects and help reduce inter-individual variability. To this end, we recruited a subset of participants from our initial study to reassess their responses to TS using the same protocol when the area of stimulation is extended from one single-digit (SD) to multi-digits (MD). Also, for this study, we elected to use only cooling stimulation, given our previous observations regarding cooling and warming effects (see above). We hypothesized that MD cooling would elicit more consistent modulation in our participants than what we observed previously with SD cooling, particularly in the form of depression, as this pattern was the most predominant in our previous report (*Ansari, Remaud & Tremblay, 2018*).

## MATERIALS AND METHODS

The study was approved by the Institutional Research Ethics Board (Bruyère Hospital Ottawa, Protocol# M16-17-001) in accordance with the principles of the Declaration of Helsinki. All participants gave written informed consent before the experimental session. All experiments were performed in a controlled laboratory environment. Participants received a small honorarium for their participation.

### Participants

As stated earlier, participants for this study were recruited from the pool ($n = 35$) who participated in our initial study (*Ansari, Remaud & Tremblay, 2018*). Since age did not affect our previous conclusions regarding TS effects, both young and senior participants were approached to participate in a second session. The final sample ($n = 22$) consisted of 13 young adults (30 ± 4 years, eight men, five women) and nine seniors (68 ± 4 years, two men, seven women). The other participants ($n = 13$) either refused or were unavailable to participate. The second experimental session took place between 2 and 9 months after the initial participation. At the times of testing, all participants were considered healthy and were free of conditions that may have interfered with the study procedures (i.e., no reports of acute or chronic musculoskeletal or neurological conditions or recent trauma to the upper extremity). In addition, they were screened to ensure that they presented no contra-indications to TMS, notably for pregnancy for young women, metallic implants in the skull and antecedents of seizures. Senior participants were also screened to ensure they were able to discriminate temperature reliably in the distal hand using tubes filled with cold (15 °C) and warm (42 °C) tap water. All but three participants (two young, one senior) were right-handed as determined by the Edinburg Hand Inventory (online version http://www.brainmapping.org/shared/Edinburgh.php).

### General procedure for TMS and recordings of motor evoked potentials

All the procedures for TMS and recordings have been described previously (*Ansari, Remaud & Tremblay, 2018*). Briefly, TMS assessments were performed in a

temperature-controlled room (22 ± 2 °C) with participants comfortably seated in a chair with armrests. MEPs elicited in the *first dorsal interosseous* (FDI, preferred hand) were recorded using surface sensors (DE-2.1; Delsys Inc., Boston, MA, USA) placed in a belly-tendon montage. After amplification and filtering (Bagnoli™ 4 System; Delsys Inc., Boston, MA, USA, bandwidth = 6–450 Hz, gain = 1,000), electromyographic signals were digitized at a rate of two kHz (PCI-63203; National Instrument Corp., Austin, TX, USA) and saved for later off-line analysis. TMS pulses were applied on the hemisphere contralateral to the preferred hand over the motor hot spot for the FDI (marked with a sticker) using a focal coil (70 mm, P/N 3190) connected to a Magstim 200 (Magstim Co. Ltd, Whitland, UK). Participants were fitted with a Waveguard™ TMS compatible EEG cap (ANT Neuro, Madison, WI, USA) with markers to ensure consistent coil placement. The resting motor threshold (rMT) was determined using the Motor Threshold Assessment Tool software (MTAT 2.0; Clinical Researcher, Knoxville, TN, USA) (*Mishory et al., 2004*). All subsequent testings were performed at 130% of the rMT with 20 trials recorded at each block. During TMS, participants were instructed to count the number of stimuli delivered to prevent shift of attention or sleepiness.

## Thermal stimulation protocol

As alluded earlier, the TS protocol included two major changes from our previous protocol. First, the extent of stimulation in the distal hand was increased to include all fingers but the thumb. Second, for this experiment, the TS consisted only of cooling stimulation for the reasons explained earlier (see Introduction). As shown in Fig. 1, the MD stimulation was obtained by applying a gel pack sleeve designed for wrist and ankle applications (TXRT-4060, 4″ × 6″; Torex® Health Products, Tallmadge, OH, USA) that covered the four fingers up to the metacarpophalangeal joint. The gel pack sleeve was of the same conductive material as the one we used for SD cooling (TXRT-2540; Torex Health Products, Tallmadge, OH, USA) in our previous experiment. Asides from these two changes, the protocol was identical to that described in *Ansari, Remaud & Tremblay (2018)*. Briefly, both skin temperature and MEPs ($n = 20$) were first measured at baseline (BL) with the fingers covered with a neutral gel pack kept at room temperature (~24 °C). Then, the cooling stimulation was applied for 5 min using a pre-cooled gel pack at ~10 °C. Such an application is in line with clinical practice guidelines for cold applications in the extremities (*Knight & Draper, 2012*) and, in our previous experiment, was effective in producing skin temperature changes in the innocuous cold range (i.e., from 15 to 25 °C). At 1 min during cooling (C1 block), both skin temperature and MEPs were measured again, which took about 1½ min to complete (i.e., 20 TMS pulses delivered with a 5 s interval between each pulse). The gel pack was kept in place until the 5 min had elapsed. Then, the cooled gel pack was removed. After 5 min post-cooling (PC5 block), skin temperature and MEPs were measured again with the hand covered with the neutral gel pack. After an additional 5 min (PC10 block), the same measures were repeated. Monitoring of skin temperature was achieved through to a K-type digital thermometer (Model# TC41FBA; Perfect-Prime, Dayton,

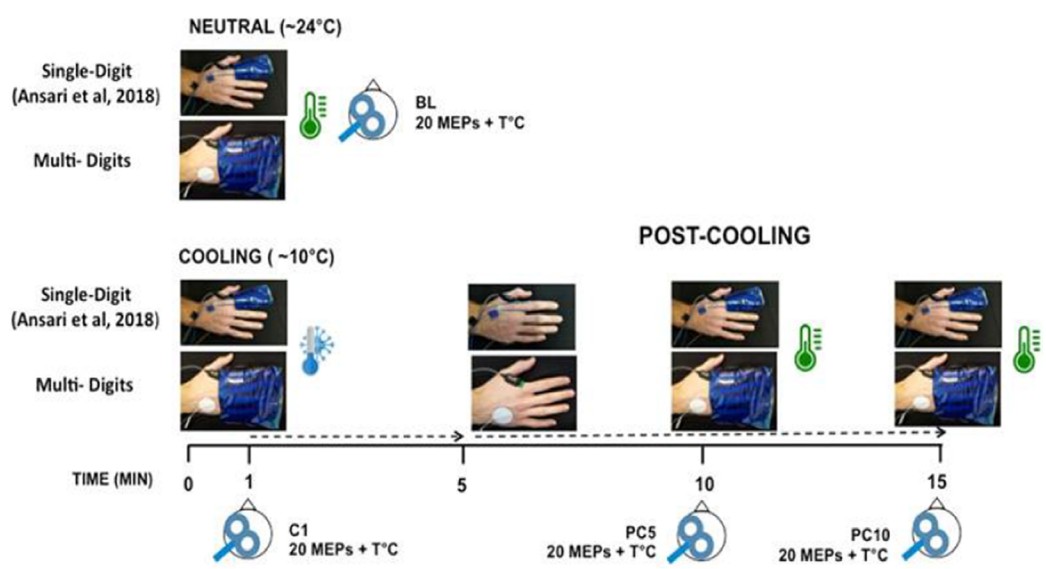

**Figure 1 Schematic representation of the experimental protocol to assess modulation in corticomotor excitability in response to distal cooling.** In our initial study, the cooling targeted a single digit (index finger) using a small gel pack sleeve. Skin temperature (T) and motor evoked potentials (MEPs) were recorded at baseline (BL, neutral gel pack), during cooling at 1 min (C1, cooled gel pack) and at 5 min (PC5) and 10 min (PC10) post-cooling with the neutral gel pack put back in place. In the current report, the thermal protocol was repeated in the same group of participants in a second testing session but this time with multi-digits cooling using a larger gel pack to cover the four fingers (no thumb) both at BL and during cooling.

NJ, USA, ±0.1 °C) connected to two thermocouple sensors affixed on the dorsal aspect of the proximal phalanx of the index and pinky fingers.

## Data analysis and statistical procedures

Individual means were obtained for BL and at the different time points in the cooling protocol by averaging recordings (two records/bock) for skin temperature and trials (20/block) for MEP amplitude (peak-to-peak) and latency. To determine the effect of stimulation area, the data obtained in our companion study with SD cooling was combined with that obtained in the current study for MD cooling in the analysis. Temperature and MEP data were checked for the normality of distribution (D'Agostino & Pearson's test) before proceeding with an analysis of variance (ANOVA). All data were normally distributed. Two-way repeated measures ANOVA's using "Area" (SD vs. MD) and "Time" (BL, C1, PC5, PC10) as the repeated factors were performed on temperature and MEP data. Only age was considered as a between-subjects factor for the present analysis, given that our previous report showed no effect of sex. Post hoc analysis was performed using the Sidak's test. Pearson's moment correlation was also used to examine relationship between variables. Additional analyses are described below. The level of significance was set at $p < 0.05$ for all tests. Statistical analyses were performed using SPSS 17.0 software package (SPSS Inc., Chicago, IL, USA) and GraphPad Prism version 7.00 for Windows (GraphPad Software, San Diego, CA, USA, www.graphpad.com). All data are reported as mean values and standard deviation.

## RESULTS

### Comparison of baseline measures of corticomotor excitability

Since the present report is based on TMS measures performed in the same group of participants at two different intervals spread over several weeks, it was critical to establish first whether BL measures of corticomotor excitability corresponded between sessions. To this end, we applied paired $t$-tests to compare rMTs and MEP characteristics. These comparisons revealed no significant difference between sessions for any of the measures. In fact, rMTs measured at session 1 (SD cooling, Mean stimulator output, 45.7 ± 12.2%) were very comparable ($t_{21} = 1.7$, $p = 0.11$) to those measured at session 2 (MD cooling, 44.1 ± 12.8%). Similarly, MEP characteristics both in terms of amplitude (Session 1, 1.8 ± 1.5 mV; Session 2, 1.9 ± 1.4 mV; $t_{21} = 0.23$, $p = 0.82$) and latency (Session 1, 23.2 ± 1.9 ms; Session 2, 23.1 ± 2.2 ms; $t_{21} = 0.64$, $p = 0.53$) showed a high degree of correspondence between sessions. Thus, BL measures of corticomotor excitability were highly comparable between SD and MD cooling testing sessions at the individual level.

### Variations in skin temperature and in MEPs in response to distal cooling: SD vs. MD stimulation

Mean skin temperature and MEP characteristics measured at BL and during the cooling protocol are compared in Fig. 2 between SD and MD stimulation. It can be seen (Fig. 2A) that skin temperature tended to be lower with MD cooling than SD cooling, although both applications exhibited a similar time course peaking at C1 (MD, −11.0 °C; SD, −10.3 °C) with a slow return toward BL in the post-cooling phase. The ANOVA revealed a large main effect of Time ($F_{3,18} = 476.7$, $p < 0.001$), but no effect or interaction with Area ($F_{1,20} = 1.83$, $p = 0.19$); indicating similar decline in temperature for both SD and MD applications. Age had no effect nor interaction ($F < 1.2$, $p > 0.36$). Post-test comparisons confirmed that skin temperature remained significantly lower than BL both during cooling (C1, $p < 0.001$) and in the post-cooling phase (PC5, $p < 0.001$; PC10, $p < 0.003$). For MEPs, it can be seen in Fig. 2B that variations in amplitude were little influenced by cooling for both SD and MD stimulation and that a substantial inter-subject variability was present at each time point. The ANOVA confirmed that neither Time ($F_{3,18} = 0.84$, $p \leq 0.50$) or Area ($F_{1,20} = 0.48$, $p = 0.50$) had an effect on MEP amplitude and also the lack of interaction ($F_{3,60} = 1.1$, $p = 0.41$). Age had no effect or interaction either ($F < 1.0$, $p > 0.54$). For MEP latency (Fig. 2C), no main effect of Time was found ($F_{3,18} = 2.3$, $p = 0.11$), although a trend was detected for Area ($F_{1,20} = 3.8$, $p = 0.06$). As shown in Fig. 2C, the latter trend reflected the difference in latency measured at PC5 for SD vs. MD cooling (23.3 vs. 24.1 ms, respectively, $p = 0.003$, Sidak's post-test). No other main effect or interaction was detected for MEP latency.

### Analysis of individual patterns of response to MD cooling

Like in our first study with SD cooling (*Ansari, Remaud & Tremblay, 2018*), closer inspection of individual responses to MD cooling revealed differences in the way participants responded to the stimulation. In fact, when individual responses were

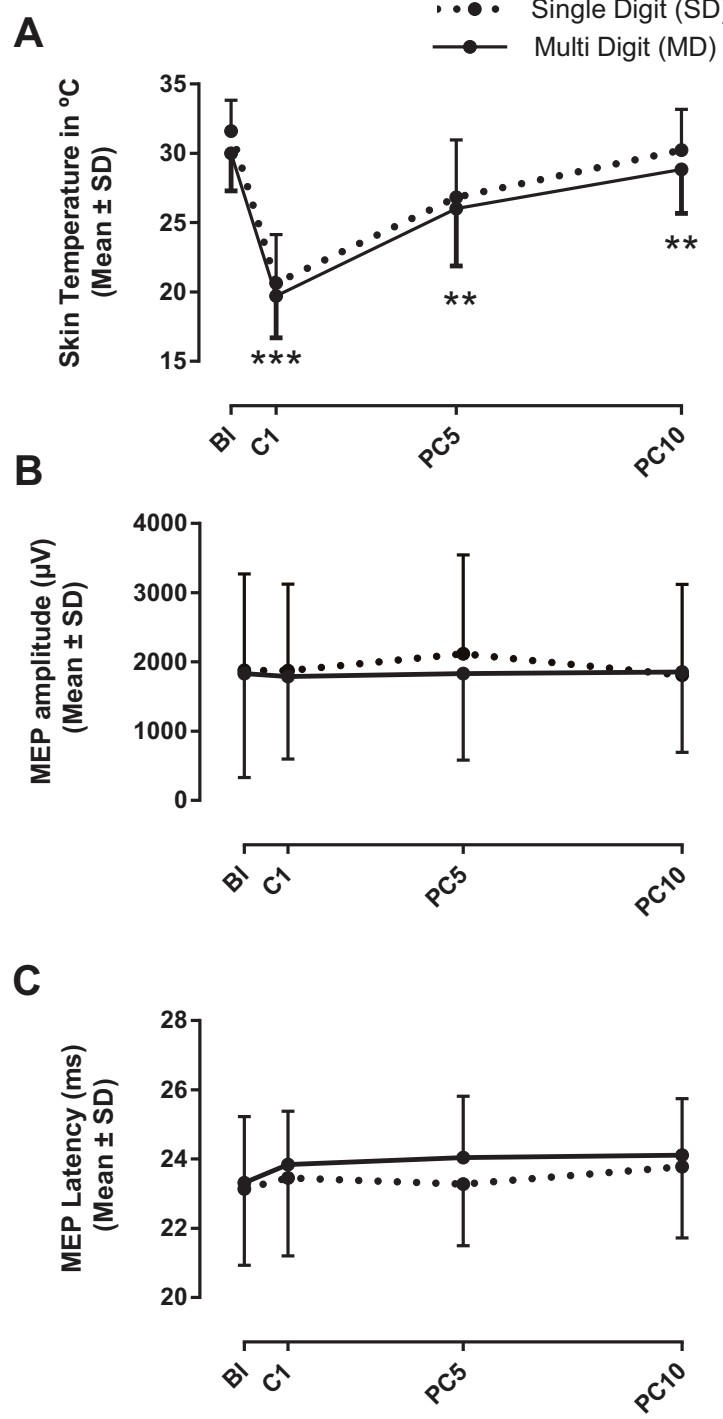

**Figure 2 Mean variations in skin temperature and in MEP amplitude and latency measured at BL and in response to cooling.** Mean skin temperature recordings and MEP data measured at BL and in response to single digit (SD) and multi-digits (MD) cooling. (A) Note the similar time course of temperature decline with SD and MD cooling with no marked difference in terms of reductions between the two. Significant differences from BL temperature were found at all time points ($^{**}p < 0.01$, $^{***}p < 0.01$). Also, note the large inter-individual variability for variations in MEP amplitude (B) and in latency (C). No effect of Area (i.e., SD vs. MD) was found for either amplitude or latency data. Abbreviations as in Fig. 1.               

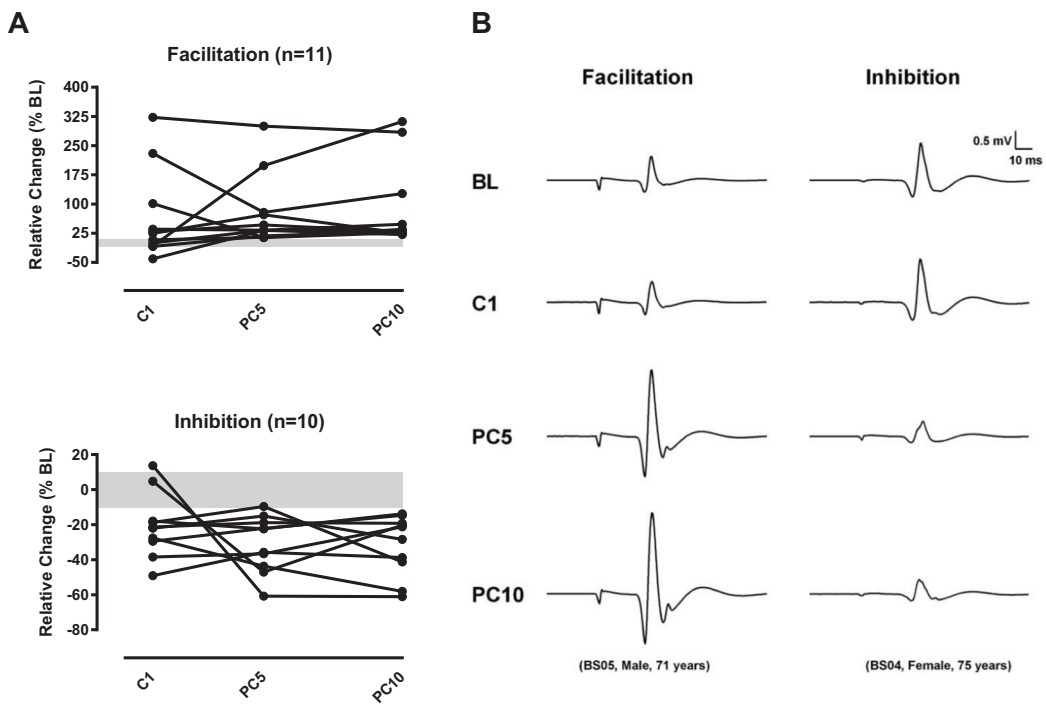

**Figure 3 Individual responses to MD cooling.** (A) Individual variations in MEP amplitude in response to MD cooling after regrouping participants according to the sign of modulation when expressed relative to percent change from BL. In each graph, the gray area represents plus or minus 10% change from BL. In half of the participants (11/22), MEPs were predominantly facilitated (i.e., MEPs > 10% BL), particularly in the post-cooling phase (PC5 and PC10). In contrast for the other half (10/22), MEPs were inhibited (i.e., MEPs < 10% BL) both during (with two exceptions) and in the post-cooling phase. One participant exhibited an inconsistent modulation. (B) Individual examples of MEP facilitation and inhibition in response to MD cooling. Note that both participants initially showed little modulation during actual cooling (C1), it is only in the post-cooling phase (PC5 and PC10) that facilitation or inhibition became evident. Abbreviations as in Figs. 1 and 2.

classified in terms of either inhibition or facilitation using a cut-off value of 10% from BL to characterize clinically relevant changes in cortical excitability (*Hinder et al., 2014*; *Perellon-Alfonso et al., 2018*), two distinct patterns emerged. As shown in Fig. 3A, for half of the participants (11/22), the pattern was characterized by predominant facilitation (MEPs > 10% BL), which was particularly evident in the post-cooling phase (i.e., PC5 and PC10). In contrast, for the other half (10/22)[1], MEPs were depressed in amplitude (MEPs < 10% BL) both during (with two exceptions) and after the cooling application. Examples of the two patterns of modulation are shown in Fig. 3B. The ratio of young/seniors and male/female was comparable in the two subgroups (Age: Facilitation, 6/5; Inhibition, 6/4; Sex: Facilitation, 5/6; Inhibition, 5/5), indicating that age and sex were not influential factors. We performed a secondary analysis on MEP amplitude data with the two subgroups of participants using one-way ANOVA with "Time" as the repeated factor. For those showing facilitation ($n = 11$), a significant main effect ($F_{3,10} = 3.9$, $p = 0.04$) was found with post-test comparisons (Dunnett's test) pointing to significant differences from BL at PC5 ($p = 0.01$) and PC10 ($p = 0.03$), but not at C1 ($p = 0.27$). Similar results were obtained for those showing inhibition ($n = 10$) with a main effect being detected for

[1] One participant showed inconsistent modulation that could not be classified as either inhibition or facilitation.

Time ($F_{3,9} = 6.7$, $p = 0.009$) and significant differences from BL being found at PC5 ($p = 0.02$) and PC10 ($p = 0.04$) in post-test comparisons. The ANOVA for variations in MEP latency showed no significant effect for the subgroup with facilitation (Time, $F_{3,10} = 1.8$, $p = 0.19$), whereas a significant effect was found for the subgroup with inhibition (Time, $F_{3,9} = 6.3$, $p = 0.006$). In this latter subgroup, post-test comparisons indicated significant differences from BL for latency measured at C1 (mean difference, +1.15 ms, $p = 0.008$) and at PC5 (mean difference, +1.25 ms, $p = 0.01$), but not at PC10 ($p = 0.10$). To summarize, participants exhibited two opposite patterns in response to MD cooling, which was characterized by a delayed effect in the form of either sustained facilitation or inhibition in the post-cooling phase.

### Correlations between modulation for SD and MD cooling application

The observation that participants exhibited variable modulation in response to cooling stimulation raised the interesting question as to whether a given individual would display a consistent response on repeated applications. To address this question, we examined the relationship between MEP modulation reported previously for SD cooling with that seen in the present report with MD cooling at the different time points. The results of this analysis are shown in Fig. 4. As evident in the Fig. 4A, during the cooling phase (C1), there was a relatively good correspondence between the modulation elicited in response to SD cooling with that observed with MD cooling. At PC5 (Fig. 4B), the correspondence was even stronger with >40% of the variance observed with MD cooling accounted for by the variance with SD cooling. At PC10 (Fig. 4C), however, the association was weaker and no longer significant.

## DISCUSSION

In the present study, we sought to extend our observations regarding thermally induced modulation in corticomotor excitability to examine the impact of stimulation area. To this end, we combined our previous observations regarding the impact of SD cooling with new ones regarding the impact of MD cooling in the same subset of participants. While our primary analysis revealed no main effect attributable to cooling area on MEP modulation, a secondary analysis indicated that participants exhibited variable responses to MD cooling, much like what we observed with SD cooling. In fact, half of the participants exhibited a predominant MEP facilitation in response to MD cooling, while the other half exhibited inhibition. This modulation in response to MD cooling was delayed and more sustained than that previously reported for SD cooling. Our results also showed that modulation in response to cooling stimulation tended to be fairly consistent at the individual level when repeated over time.

### Measures or resting corticomotor excitability at BL

As stated earlier, one critical aspect of the present report was to ensure that participants exhibited comparable excitability measures at BL. This was indeed critical, given the longitudinal design of the study, where previous measurements derived from a subset of participants were compared to new measurements obtained several weeks

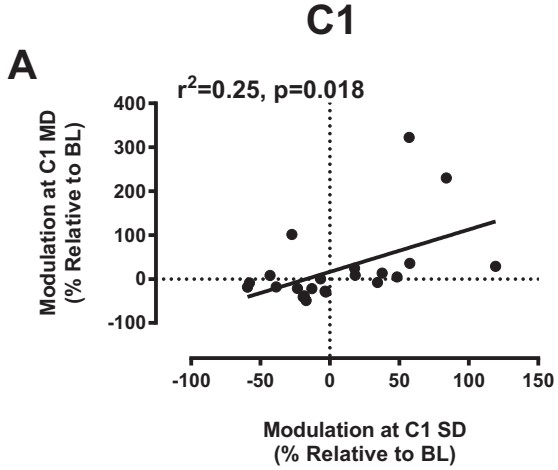

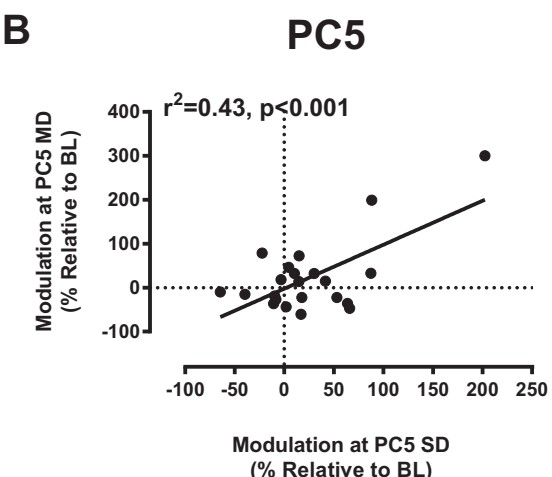

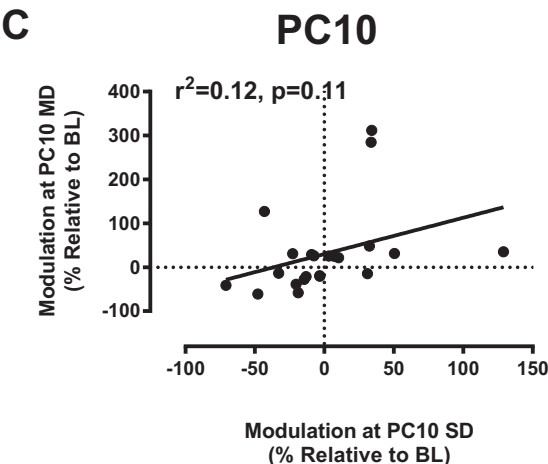

**Figure 4 Correlations between MEP modulation with MD and SD cooling.** Correlations between MEP modulation observed in the second session for MD application with that observed in the first session for SD application at each time point relative to cooling stimulation. Note the relatively good degree of correspondence between sessions in (A) and (B) for MEP modulation elicited at C1 and PC5, but not at PC10 (C). Abbreviations as in Figs. 1 and 2.
after the initial experiment. Our design was justified given the evidence that basic measures of excitability, such as rMT and MEPs at suprathreshold intensity, are relatively stable over time for a given individual (*Brown et al., 2017*). In line with this, our basic measures of resting corticomotor excitability (rMT and MEP) data showed very good reproducibility over time in our group of participants; which strengthens our contention that MEP modulation observed in the present report reflected physiological responses to cooling stimulation and not just random fluctuations in excitability.

## Variations in skin temperature and in MEP amplitude in response to cooling

Regarding skin temperature, we observed a similar profile of declining temperature with MD cooling as the one we observed with SD cooling. This was expected given that thermal agent was made of the same conductive material and was applied at the same temperature to restrict cooling to the innocuous range. The observation that the range of temperatures recorded at C1 was comparable for both SD and MD cooling (range, 13.0–26.6 and 13.3–26.0, respectively) confirms that both applications produced the desired decrease in skin temperature. Although we did not record subjective ratings, most participants reported that their sensory experience with the larger gel pack (MD cooling) was more intense (i.e., colder) than that felt previously for the smaller gel pack (SD cooling); an observation consistent with the effect of spatial summation on perceived cold sensation (*Stevens & Marks, 1979*).

With regards to MEP modulation, our primary analysis revealed no effect attributable to area, a finding that goes against our prediction that increasing stimulation area would produce more consistent effects and help reduce variability. Such a conclusion, however, would obscure the fact that almost all participants exhibited a significant modulation in response to MD cooling, and only that this modulation was of opposite signs for half of them. In fact, similar to our previous observations with SD cooling, the presence of large subsets of participants showing either inhibition or facilitation contributed to blur any effects attributable to increased stimulation area. We have discussed previously the possible reasons as to why corticomotor excitability could be depressed in one individual and enhanced in another in response to the same cooling stimulation applied peripherally (see *Ansari, Remaud & Tremblay, 2018*). Besides the inherent variability of individual responses to sensory stimulation reported in studies examining modulation of excitability (*Chipchase, Schabrun & Hodges, 2011*), we can reiterate the possible role of spinal mechanisms to explain the presence of facilitation in many participants, as local cooling in the extremities is known to increase motoneuronal excitability (*Dewhurst et al., 2005*; *Palmieri-Smith et al., 2007*). For those showing inhibition, as we have argued before (*Ansari, Remaud & Tremblay, 2018*), spinal facilitation may have been overrun by inhibition exerted at the motor cortical level via activation of somatosensory areas (primary and secondary) and insular cortex resulting from cold afferent stimulation (*Casey et al., 1996*; *Craig et al., 1996*). The fact that the MEP latency was significantly delayed in the subset with inhibition, but not with those with facilitation, would be compatible with a depressed excitability at the cortical level

since prolonged latency could reflect a reduced excitatory drive to corticospinal neurons leading to greater temporal dispersion of descending impulses. Alternatively, a temperature-dependent decrease in peripheral nerve conduction is also possible to account for the prolonged MEP latency, although this explanation is hardly compatible with the observation that latency was changed in only one subset (those with inhibition) and only for certain time points (i.e., C1 and PC5). Besides, we have provided evidence previously (*Ansari, Remaud & Tremblay, 2018*) that cooling restricted to the distal finger did not affect proximal nerve conduction. Ultimately, the presence of differential responses to cooling may reflect individual differences in the way thermal afferent information is processed from the periphery (e.g., differences in skin properties, and in receptors density) up to the cortex (e.g., degree and extent of cortical activation).

Whatever the reasons for the presence of facilitation and inhibition in response to cooling, one noticeable difference between SD and MD stimulation was in the time course of the modulation. With SD cooling, the modulation was restricted mainly to the cooling phase (i.e., C1), whereas with MD cooling, the modulation was delayed to the post-cooling phase (i.e., PC5-PC10) for both facilitation and inhibition. Such a difference in modulation could be related to factors linked with spatial summation of cold afferent and the way the cooling agents interacted with the skin locally. The observation that MEP modulation was delayed with MD cooling might have reflected differences in the efficiency of the cooling application, which could not be detected with our temperature sensors. For instance, the large gel pack covered only the dorsal and palmar aspects of the index finger and not its lateral aspects, unlike the small gel sleeve we used for SD cooling. Thus, initially the cooling effects might have been more efficient with the small gel sleeve than the large gel sleeve, as far as the index finger (and FDI muscle) is concerned; hence the lack of clear modulation at C1. However, as time passed, the rapid increase in the number of active cold afferents and their activity level, as the cooling extended spatially to adjacent fingers, might have contributed to sustain the afferent-induced modulation for minutes, even after the gel pack had been removed. Summarizing, while our main analysis failed to confirm our predictions regarding the influence of stimulation area, a secondary analysis of individual responses provided evidence that increasing cooling area was associated with a more sustained modulation in the form of either inhibition or facilitation.

## Correlations between SD vs. MD cooling

Considering the variability of individual responses to cooling stimulation, it was important in the present report to address the issue of repeatability, that is, whether a given individual would show consistent responses on repeated applications. In this regard, our correlative analysis of MEP modulation elicited with either SD or MD cooling provides some interesting insights on this important question. In particular, our correlations showed that modulation elicited in a previous session with SD cooling were significantly associated with those elicited with MD cooling in the current study so that individuals that showed inhibition with SD cooling also tended to show inhibition with MD cooling (same for those with facilitation). This association was particularly strong in the

post-cooling phase at PC5, where >40% of the variance with MD cooling was explained by SD cooling; an observation that can be linked with the delayed modulation associated with the larger gel pack, as discussed in the preceding section. Along the same line, the lack of significant association at PC10 could also be explained by the observation that modulation was more sustained with MD, as compared to, SD cooling. As stressed earlier, while the reasons as to why someone would show facilitation and another one inhibition remain unclear, our results nevertheless show a fairly good probability that an individual showing depressed (or enhanced) excitability in response to cooling would also showed a depression (or facilitation) over time with repeated applications.

## CONCLUSIONS

The present results extend our previous observations regarding the influence of distal local cooling stimulation on corticomotor excitability. In particular, our observations show that increasing the cooling area in the distal hand, while still eliciting variable responses in terms of facilitation and inhibition, is associated with more sustained modulation in the post-cooling phase. In addition, our correlative analysis of MEP modulation between sessions for SD and MD cooling provides evidence of a fairly good within-subject repeatability over time on repeated applications. While our observations were obtained from healthy participants, they nevertheless point to critical aspects regarding the physiological effects of cooling stimulation as a means to modulate corticomotor excitability for rehabilitation purposes.

## ACKNOWLEDGEMENTS

The authors wish to thank all participants for their time and patience during testing. Part of this work was performed in the context of a PhD in Rehabilitation Sciences by Yekta Ansari.

### Funding

Y. Ansari received financial support from the Faculty of Health Sciences, University of Ottawa, in the form of a graduate scholarship. This research was also supported by the Graduate Studentship Program at the Bruyère Research Institute. The funders had no role in study design, data collection and analysis, decision to publish, or preparation of the manuscript.

### Grant Disclosures

The following grant information was disclosed by the authors:
Faculty of Health Sciences, University of Ottawa, in the form of a graduate scholarship.
Graduate Studentship Program at the Bruyère Research Institute.

### Competing Interests

The authors declare that they have no competing interests.

## Author Contributions

- Yekta Ansari performed the experiments, analyzed the data, contributed reagents/materials/analysis tools, prepared figures and/or tables, authored or reviewed drafts of the paper, approved the final draft.
- Anthony Remaud performed the experiments, analyzed the data, contributed reagents/materials/analysis tools, approved the final draft.
- François Tremblay conceived and designed the experiments, performed the experiments, authored or reviewed drafts of the paper, approved the final draft.

## Human Ethics

The following information was supplied relating to ethical approvals (i.e., approving body and any reference numbers):

The Bruyère Research Ethics Board approved the study (Protocol #M16-17-00).

## Data Availability

The raw measurements are provided in the Supplementary File.

## Supplemental Information

Supplemental information for this article can be found online at http://dx.doi.org/10.7717/peerj.6163#supplemental-information.

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
