# Peer review of "Modulation of corticomotor excitability in response to distal focal cooling"

_PeerJ, doi:10.7717/peerj.6163_

## Round 0.1 · original submission · Minor Revisions

The reviewers commented positively on the manuscript and made several comments that need to be addressed before the manuscript can be accepted for publication.

Reviewer 1 ·

Basic reporting

Line 45-60: The authors described “there is still very limited information regarding the neural basis underlying such facilitatory effects.”
The introduction attempted to give a background on this study but overall, the descriptions on neuropathway from sensory stimulation receptor to motor cortex are lacking. In addition, a definition of what is meant by thermal stimulation would be good and the range of temperatures of thermal stimulation should be described.

Experimental design

Method
Line 97-99: “As described previously, all participants were considered healthy and were screened to ensure that their sensations were normal and that they had no contra-indications to TMS.”
I think it would be better list the inclusive and exclusive criteria items.

Line 129-135: How many minutes to spend in completing TMS testing during thermal stimulation protocol? Please clarify the short intervals between 1-min and 5- and 10-min post-cooling during the cooling application whether enough time to measure TMS parameters.
In addition, the authors did not state any past experience regarding safety of the procedures. Has the safety issue been studied? It would be better to clarify the reason using a pre-cooled gel pack at~10 °C.

Validity of the findings

Results
Line193-195: It would be better to add the citation reference for the statement “In fact, when individual responses were classified in terms of either inhibition (MEP<10% BL), facilitation (MEP>10% BL) or no modulation (MEP ±10% BL), two distinct patterns emerged”

Discussion
Regarding the resting motor threshold (rMTs), MEP amplitude and latency, the authors should state the clinical implication for these parameters.
The description in discussion section is too long. Can you make it more succinct? However, it would be helpful to address the limitation section in the revised manuscript.

Conclusion
Line 339-341: “Such observations, notably with regards to spatial summation effects and repeatability between sessions, seem critical for future therapeutic applications of cooling stimuli for rehabilitative purposes in neurological populations.”
Only healthy adults (n=22) were recruited in this study and thus the results of the present study cannot support the above-mentioned statement. I suggest the authors delete the statement.

Additional comments

This cross-sectional study is an interesting and practical research project. The research entitled “Modulation of corticomotor excitability in response to distal focal cooling”. The aim of this study was to examine whether an increase in stimulation area from single-digit (SD) to multi-digit (MD), four fingers could elicit more consistent modulation. Skin temperature and motor evoked potentials (MEPs) elicited with transcranial magnetic stimulation were measured at baseline (BL, neutral gel pack at 22 ◦C), at 1-min during the cooling application (pre-cooled 10◦C gel pack) and 5- and 10-min post-cooling (PC5 and PC10). The results highlight that responses to cooling stimulation in terms of either depression or facilitation of corticomotor excitability tend to be fairly consistent in a given individual with repeated applications. However, the generalizability of the results with regards to spatial summation effects and repeatability between sessions for rehabilitative applications in neurological populations were limited.

·

Basic reporting

The study addressed the impact in thermal stimulation has been proposed to facilitate motor recovery in neurological populations, such as stroke. The authors extended their own previous work with single digit cooling, and extended it to multiple digit cooling based on their goal of determining whether an increase in stimulation area could elicit more consistent modulation of motor evoked potentials elicited by Transcranial magnetic stimulation (TMS) directed to the motor area of the first dorsal interosseous (FDI) muscle.
The study is clearly written and the English is of a high standard. The literature review is well referenced and contextualizes the background to the study well. The figures are clear and well labelled and the raw data is supplied.

Experimental design

The authors extended their own previous work with single digit cooling, and extended it to multiple digit cooling to address their stated research question of determining whether an increase in stimulation area could elicit more consistent modulation of motor evoked potentials elicited by Transcranial magnetic stimulation (TMS) directed to the motor area of the first dorsal interosseous (FDI) muscle. The research fills an important gap about the possible mechanisms by which cool influences cortical excitability.
The methods are described in detail and important additional measures such as skin temperature to confirm that cooling indeed took places were included. The work was performed to a high technical and ethical standard.
The overall group results were not significant but the authors observed that half of participants showed post cooling MEP facilitation while the other half showed post cooling MEP depression. Further they showed that the tendency to either inhibit or depress MEPs in response to cooling was consistent with participants who participated in their previous single digit cooling experiment and the current multi-digit cooling.
This is the biggest strength of the work. There is a growing trend to determine if MEP responses to interventions show a differential response to a given intervention, this work clearly does.

Their conclusions were fair, related to the original research question and did not overstep the mark. Their correlative analysis of MEP modulation between sessions for SD and MD cooling found reasonable within-subject repeatability over time on repeated applications of cooling. This suggests that the tendency for MEPs to either inhibit or facilitate in response to cooling is consistent within a given individual, providing validity to their decision to report those who facilitated and those who inhibited in response to cold as two separate groups. Their conclusion that this work on the impact of spatial summation effects and individual repeatability between sessions, is important work needed to validate “future therapeutic applications of cooling stimuli for rehabilitative purposes in neurological populations.”
Suggestion: if possible, include 2 or 3 references from previous TMS work where a bimodal response in terms of MEP inhibition or facilitation has been observed. This is not essential but it would strengthen their decision to compare the “inhibitors” and “facilitators” as two separate groups.

Validity of the findings

The study has in important impact because thermal stimulation has been proposed to facilitate motor recovery in neurological populations, such as stroke, and yet little is understood regarding the physiological mechanisms by which thermal stimulation impacts neural function. In this study, neural function was assessed by measuring changes in motor evoked potential (MEP) amplitude and latency at varying time points during and after multi-digit cooling. The data is robust (averaging 20 MEPs at each time point), and the statistics are appropriate for the research design.

---

## Round 0.2 · accepted · Accept

The authors have adequately addressed the reviewer comments

# Reviewer 1 ·

Basic reporting

The author has made an appropriate response.

Experimental design

The author has made a satisfactory revision for the comment.

Validity of the findings

The author has made an appropriate response.

Additional comments

The author has made an appropriate and satisfactory response.